# Evaluation of Bispecific T-Cell Engagers Targeting Murine Cytomegalovirus

**DOI:** 10.3390/v16060869

**Published:** 2024-05-29

**Authors:** Hanna Menschikowski, Christopher Bednar, Sabrina Kübel, Manuel Hermann, Larissa Bauer, Marco Thomas, Arne Cordsmeier, Armin Ensser

**Affiliations:** Institute of Clinical and Molecular Virology, University Hospital Erlangen, Friedrich-Alexander-Universität Erlangen-Nürnberg (FAU), 91054 Erlangen, Germany; hanna-menschikowski@t-online.de (H.M.); sabrina.kuebel@uk-erlangen.de (S.K.); manuel.hermann@fau.de (M.H.); arne.cordsmeier@uk-erlangen.de (A.C.)

**Keywords:** cytomegalovirus, murine cytomegalovirus, CMV, BiTE, bispecific T-cell engager, T cell, T-cell toxicity, bispecific

## Abstract

Human cytomegalovirus is a ubiquitous herpesvirus that, while latent in most individuals, poses a great risk to immunocompromised patients. In contrast to directly acting traditional antiviral drugs, such as ganciclovir, we aim to emulate a physiological infection control using T cells. For this, we constructed several bispecific T-cell engager (BiTE) constructs targeting different viral glycoproteins of the murine cytomegalovirus and evaluated them in vitro for their efficacy. To isolate the target specific effect without viral immune evasion, we established stable reporter cell lines expressing the viral target glycoprotein B, and the glycoprotein complexes gN-gM and gH-gL, as well as nano-luciferase (nLuc). First, we evaluated binding capacities using flow cytometry and established killing assays, measuring nLuc-release upon cell lysis. All BiTE constructs proved to be functional mediators for T-cell recruitment and will allow a proof of concept for this treatment option. This might pave the way for strikingly safer immunosuppression in vulnerable patient groups.

## 1. Introduction

In the advent of new immunosuppressive therapies, particularly in the context of transplantation, clinicians witness a rise in infectious complications. One of the main adversaries is the human cytomegalovirus (CMV) [1], a ubiquitous beta-herpesvirus with population-dependent seroprevalence of up to 90% [2] that can cause severe systemic inflammation and organ failure with a lethal outcome [3,4]. The current therapy options include antivirals such as (Val-) Ganciclovir [5,6,7,8] or Letermovir [9,10], and, recently, Maribavir, which are CMV- and herpes-virus-specific, respectively. They can be applied as a universal prophylaxis for patients at risk, or are used within pre-emptive prophylaxis regimes [11], as soon as a set viremia threshold is crossed [4,12]. However, adverse reactions, side effects, and emerging drug resistance complicate treatment [8].

Another novel strategy to fight CMV disease is the ex vivo expansion and adoptive transfer of CMV-specific T cells. This is an expensive, time-intensive, and technically sophisticated process [13]. However, patients need a readily available, off-the-shelf treatment option, especially in a high-risk constellation like a CMV-negative donor/CMV-positive recipient hematopoietic stem cell transplant (HSCT) [14,15,16,17]. These patients would benefit from bridging therapy until the transplanted immune system can mount a sufficient response to infection or reactivation. Transplant engraftment takes up to a month, and, while infections occur during cytopenia and after, this does not allow sufficient time to develop an adaptive immune response [18]. Therefore, physicians observe a vulnerable phase of considerable length. Herpes virus eradication is neither the aim nor feasible for these patients, but, rather, the suppression of active replication is required.

Previous studies have shown that T cells are pivotal in controlling CMV infection and maintaining the latent state. Therefore, we concluded that it might be advantageous to support the inherent abilities of adaptive immunity in a near-physiological state. The impact of CD8+ T cells in infection control have been long-established [19] and recent studies show that CD4+ T cells also play a major role in long-term protection [20,21,22,23,24,25]. In this respect, we built on an established concept from oncology, the bispecific T-cell engager (BiTE). Blinatumomab is a CD19-targeting, licensed BiTE immunotherapy developed by Amgen Munich (formerly Micromet Inc.) [26]. BiTE proteins are designed to connect the immunological synapse of effector T cells to the target protein and induce target cell death (Figure 1B). Notably, this is independent from classical T-cell recognition via the major histocompatibility complex (MHC) presentation and, thus, resistant against MHC-related immune evasion mechanisms of tumour cells or virus-infected cells. The main advantage of MHC independence, however, is the option for a non-individualised, off-the-shelf treatment. While CMV inherently exploits several immune evasion mechanisms, the antagonisation of the MHC-dependent immune response is central and conserved among the cytomegaloviruses. Hence, with the application of BiTE, T-cell activation relies on the interaction of CD3ε on the T cell and the specific target protein connected by BiTE, a mechanism likely not encountered by cytomegaloviruses throughout evolution, and, thus, will allow the T-cell immune recognition of infected cells.

BiTE consist of two single-chain variable fragments (scFv) derived from antibodies that are connected by a short linker sequence (Figure 1A). This allows a protein mass of only 55 kDa for the better control of adverse reactions and tissue permeability, but is accompanied by the drawback of a short half-life [27].

In contrast to individualised therapies, BiTE provide a manufacturing process with easy storage for clinical application, similar to monoclonal antibodies or other biologicals. Additionally, interactions with classic drug metabolism pathways, such as Cytochrome P450 3A4 (CYP3A4), are not expected and, therefore, allow the safer combination with traditional drugs [11]. Highly promiscuous metabolisation enzymes such as CYP3A4 interact with up to 50% of the available drugs including antibiotics and immunosuppressants. Both drug groups are essential parts of transplant regimes and depend on sufficient drug levels [28,29].

We designed BiTE constructs to target murine cytomegalovirus (MCMV) in order to prove the concept of antiviral BiTE efficacy in vitro and in vivo. We chose the mouse model since it is well-established [30,31] and the accepted small-animal model for the highly species-specific human CMV [32,33]. In future studies, immunodepleted mice will be infected via MCMV and treated by BiTE with murine anti-CD3ε domain injected together with naïve donor effector T cells. For this study, we used human anti-CD3ε domain from Blinatumomab to circumvent issues regarding the anti-CD3 domain and to focus solely on our novel binder. As this is the baseline qualitative study, we use MCMV as a proof of concept for efficacy testing. However, our group has previously shown the translatable evidence of function with human CMV targeting the BiTE [34]. However, to evaluate this in vivo in a model of systemic infection, we would need a recombinant MCMV expressing human CMV glycoproteins. They could then be tested either with murine or human anti-CD3 domain, which would both not be as physiological as using murine virus and effector cells. An actual bench-to-bedside translation is only possible when we evaluated the closest set up to a physiological system.

Cytomegaloviruses have evolved very sophisticated immune evasion mechanisms. Viral effector mechanisms in infected cells interfere with various steps of antigen presentation and the immune response, for instance, by inhibiting natural killer (NK) cell activation [35]. Notably, NK cells are unable to kill CMV-infected cells in vitro but still restrict the virus by Interferon response [36,37]. Similarly, in MCMV infection, NK cell activation relies heavily on CD4+-mediated Interferon γ release, hinting at a strong contribution of spread inhibition rather than killing. It is important to note that, especially in MCMV, there is a high strain variability in the NK response depending on the Ly47+/− status and MCMV m157 binding [38,39]. In latency, herpesvirus-infected cells are almost invisible to the host immune response and express no viral proteins suitable as a drug target, making eradication practically impossible. During active replication, however, highly conserved glycoproteins that are necessary for viral spread and entry are expressed on the cell surface [40]. Thus, we consider them key targets for our study. The glycoprotein B (gB) is well-established in the literature and has been proven as a promising target [41]. We [34] and other research groups have successfully targeted gB with bispecific constructs or antibodies [42] in both, CMV, and MCMV. The further two glycoproteins N (gN) and H (gH) investigated in this study are additional promising targets and have been addressed in antibody experiments [43,44], and, thus, are great candidates for BiTE experiments [43,44].

As MCMV target homologues, we chose MCMV M55 for gB, M73 for gN, and M75 for gH. They have been established as positional and functionality homologues to their human CMV counterparts.

While few studies have been carried out about the gN-induced entry mechanism, we know of its susceptibility as a neutralisation target and its role in immune evasion [43,45]. It is expressed in a complex with glycoprotein M (gM). Together, gN and gM form the most abundant glycoprotein complex on HCMV virions [46]; they are conserved across the herpes virus family—however, not to the extent that cross-recognition by adaptive immune cells can be expected [43]. Glycoproteins gH and L (gL), as part of a complex, bind to platelet-derived growth factor receptor alpha (PDGFR-α) and allow viral membrane fusion and entry into the cell. In contrast to gB, which is responsible for viral fusion, this complex is required for cell-to-cell spread and allows spread within tissue [47].

Together, these three viral surface glycoproteins represent a broad spectrum of unrelated targets that may be addressed to inhibit viral spread. Both gH and gN, as part of the gHgL and gNgM complex, have not been targeted by BiTE before and may provide an opportunity for combination therapy against multiple targets to inhibit the viral replication.

Due to a large variety of CMV immune escape mechanism complicating the experimental readout [48], we implemented stable cell lines to express the viral glycoproteins for target validation. By subtracting viral immunoediting and T-cell escape, we can isolate the sole impact of our constructs and, therefore, show a proof of function for our new BiTE in vitro. They were successfully secreted by eukaryotic cells and directed T-cell effector functions to kill MCMV-glycoprotein-expressing target cells. Especially for CMV gNgM, this has not been shown previously.

All constructs proved to be viable options for further characterisation. Altogether, this now forms the base for the future in vivo characterisation in the MCMV mouse model.

## 2. Materials and Methods

### 2.1. BiTE Plasmids

All constructs are based on a pCDNA4 vector (Invitrogen. Carlsbad, CA, USA) with an in-frame fusion to a N-terminal Flag-Tag. The human CD3ε-binding domain sequence was modelled from Blinatumomab [49]. This, combined with the virus-specific scFv [50] encoding gene blocks (Integrated DNA Technologies, Coralville, ID, USA), were inserted via enzyme digestion and Gibson assembly [51]. Virus-specific binding sites are derived from sequences of monoclonal antibodies kindly provided by the laboratory of Michael Mach, UK Erlangen and Thomas Winkler, FAU Erlangen [52]. Here, the sequence of the variable region of heavy and light chains was connected to the CD3ε-binding domain via a glycine–serine–glycine (GSG) linker. The sequence of the six generated BiTE constructs (Table 1) was confirmed via Sanger sequencing (Macrogen Europe, Amsterdam, the Netherlands).

### 2.2. Plasmids for Generation of Reporter Cells

Lentiviral vector subcloning was carried out in pLV-EF1α-IRES-Hygro (#85134), pLV-EF1α-IRES-Puro (#85132), and pLV-EF1α-IRES-Blast (#85133) obtained from Addgene (Watertown, MA, USA). Expression vectors for MCMV glycoproteins were generated using Gibson assembly (Table 2, sequences provided by Marco Thomas, UK Erlangen). Glycoprotein N and H are coexpressed with gM and gL, respectively (Figure 1), imitating their physiological complexes for surface expression.

All glycoprotein-presenting cells additionally express nano-luciferase (nLuc) as reporter to enable readout of killing assays (Figure 2). Plasmid DNA was isolated from *E. coli* using the PureLink™HiPure Plasmid Midiprep Kit (Thermo Fisher Scientific, Waltham, MA, USA).

### 2.3. Cell Culture

#### 2.3.1. HEK293T Reporter Cell Lines

All cells were cultivated at 37 °C, 5% CO_2_, and 80% humidity. Human embryonic kidney (HEK) 293T cells (ATCC^®^ CRL-3216 ™, ATCC, Manassas, VA, USA) were cultivated in Dulbecco’s Modified Eagle Medium (DMEM; 11500516, Thermo Fisher Scientific, Waltham, MA, USA) supplemented with 10% heat-inactivated fetal bovine serum (FBS; 10270106, Thermo Fisher Scientific, Waltham, MA, USA), 50 μg/mL gentamicin (1405-41-0, Serva Electrophoresis, Heidelberg, Germany), 2 mM GlutaMAX™ (35050061, Thermo Fisher Scientific), and 25 mM HEPES (15630080, Thermo Fisher Scientific).

Glycoprotein-expressing cells were selected by supplementation with antibiotics 5 μg/mL blasticidin (ANT-BL-1, InvivoGen, San Diego, CA, USA), 2 μg/mL puromycin (C-1080713, Santa Cruz Biotechnology, Dallas, TX, USA), and 200 μg/mL hygromycin B (H0654, Merck), respectively, as shown in Table 2.

#### 2.3.2. CB15 T Cells

CB15 is a primary human CD4+ T-cell line immortalised by Herpesvirus saimiri strain C488 [53]; CB15 can serve as standard cytotoxic effector cells and are more convenient to cultivate and more viable than, for example, their CD8+ 3C counterpart [54]. They were kindly provided by Brigitte Biesinger-Zwosta, UK Erlangen. We cultivated them in Roswell Park Memorial Institute 1640 medium (RPMI 1640; 11875093, Thermo Fisher Scientific) with 50 U/mL recombinant human (rh) Interleukin 2 (IL-2; 11147528001, Merck, Darmstadt, Germany) and supplemented with 10% heat-inactivated FBS, 50 μg/mL gentamicin, 2 mM GlutaMAX™, and 25 mM HEPES.

### 2.4. Transfection

For transfection, polyethylenimine (PEI; 23966-100, Polysciences Inc, Warrington, PA, USA; 1 mg/mL) was used in a 3:1 (*w*/*w*) ratio with DNA in amounts appropriate for flask surface area. HEK293T cells were seeded a day prior to transfection, and then the appropriate amount of PEI and DNA were diluted in reaction tubes, rested to allow formation of PEI-DNA complexes, added to cells, and incubated for 4–6 h before medium exchange.

### 2.5. Lentivirus Production

Lentiviral plasmids (Table 2) were transfected in a 3:1:1 ratio mix with helper plasmids psPAX2 and pMD2.G (kindly provided by Prof. Didier Trono, École Polytechnique Fédérale de Lausanne, France) using the protocol for PEI transfection. Supernatant containing lentivirus was harvested after 3–5 days, separated from cell debris using centrifugation (300× *g*, 5 min) and used for transduction immediately or stored at −80 °C.

### 2.6. Transduction

Using T25 flasks, HEK293T cells were seeded one day prior to transduction and 3 mL of lentiviral supernatant were added. Cells were incubated for 2–3 days before antibiotics were supplemented (Table 2) and selection observed for 1–2 weeks.

### 2.7. Generation of BiTE

BiTE plasmids were transfected in HEK293T cells using the PEI protocol described above and rested for 10 days. HEK293T cell supernatant was harvested, centrifuged to remove cell debris (300× *g*, 5 min), and stored at 4 °C under sterile conditions.

### 2.8. Flow Cytometry

HEK293T MCMV-glycoprotein-expressing cells were resuspended and harvested, and then washed in FACS buffer (5% FBS in phosphate-buffered saline, PBS). If BiTE staining was intended, BiTE supernatant was added and incubated for 1 h, and then washed twice with FACS buffer. Primary antibody was diluted according to manufacturer’s protocol (see Table 3) and cells were incubated for another hour in 100 µL of the respective dilution or hybridoma supernatant. After washing again, secondary antibody was diluted according to manufacturer’s protocol and incubated for another hour. With two consecutive washings, cells were analysed using an Attune^®^ NxT Acoustic Focusing Cytometer (Thermofisher Scientific) and FlowJo™ v10.8 Software (BD Life Sciences, Franklin Lakes, NJ, USA) version 10.8.1 (Becton, Dickinson and Company, Ashland, OR, USA). For intracellular staining, cells were incubated in FACS buffer supplemented with 0.5% Saponin.

### 2.9. CB15 T-Cell-Induced Killing: nLuc Assay

HEK293T MCMV-glycoprotein-expressing target cells were seeded on day 0 without selection antibiotics in a flat-bottom 96-well plate (10,000/well). On day 1, 100 µL of the respective BiTE supernatant and 50,000 CB15 effector T cells per well were added. On the next day, 50 µL supernatant were analysed for nLuc activity as correlate for target cell lysis using Nano-Glo^®^ Luciferase Assay System (N1120, Promega Madison, WI, USA) and the ORION II Luminometer (Berthold Technologies GmbH & Co. KG, Bad Wildbad, Germany) as depicted in Figure 3.

## 3. Results

### 3.1. Design of Bispecific T-Cell Engager (BiTE)

Carefully weighing useful targets for our BiTE, we chose among several monoclonal antibodies against MCMV glycoproteins which provided the sequences for the viral protein targeting scFv (Figure 1A, green) which are linked via the GSG linker to the human CD3ε-recognising single-chain fragment (Figure 1A, blue). For secretion, a signal peptide was added to the N-terminus, as well as Flag and His epitope tags for detection (Figure 4). As control, we adapted the sequence of Blinatumomab targeting CD19 for human CD3ε [26].

### 3.2. BiTE Expression and Binding Ability

Since BiTE have a molecular mass of approximately 55 kDa, in roughly the same order as highly abundant serum proteins like albumin, they are hard to detect in culture supernatant even using their tags. For that reason, we aimed to prove the expression and binding in one. For our model system, we transduced target proteins’ expression vectors into HEK293T cells. Every protein-coding vector encodes a different antibiotic resistance; therefore, higher expression levels can be maintained under selective pressure. Furthermore, selection pressure allows the verification of the expression before conducting each assay. Glycoprotein targets gH and gN rely on gL and gM, respectively, for efficient surface presentation [55,56]. This was taken into consideration as seen in Figure 1.

#### 3.2.1. Glycoprotein Expression

In order to assess the transduction and expression of target proteins, we utilised flow cytometry with hybridoma supernatants (kindly provided by Larissa Bauer, UK Erlangen). We were able to detect gH at 89.5%, gB at 95.3%, and CD19 at 99.6% at the surface of cells. Due to the instability of the utilised gN hybridoma supernatant, gN was detected by the intracellular HA-tag at 96.4% of the cells. The cell surface expression was evaluated before each target cell for the nLuc-release assay and cells were constantly held under Darwinian selection to uphold glycoprotein expression (Figure 5).

#### 3.2.2. BiTE Binding Ability

Having proven the glycoprotein expression, we proceeded to use the respective BiTE (nomenclature shown in Table 1) similar to an antibody to investigate their binding abilities. For the detection glycoproteins, BiTEs were stained using anti-Flag antibody to measure if the protein was bound to the target glycoprotein on the cells. BiTEs binding properties are visualised in Figure 6.

In general, all BiTE bound to their target, visible by an increase in fluorescence compared to HEK293T cells alone (Figure 6, lower panels) as well as HEK293T cells expressing the respective glycoprotein but not treated with BiTE-containing supernatant (Figure 6, middle panels). Particularly, CD19human proved to be high in binding, validating itself as a relevant control and demonstrating the functionality for this assay. When looking at 32.22human, we see a high background signal (Figure 6), which is explained by the gM construct also possessing a Flag-tag. However, when BiTE 32.22human is added, there is a clearly visible shift, proving the BiTE binding capacity.

While we cannot compare between BiTEs due to the unknown concentration, we can examine all of them individually. With all BiTEs showing a detectable expression and binding, we provided a valuable basis to move on to killing assays. Additionally, this is the first step in confirming new potential targets gH and gN for antiviral T-cell therapy, adding to the current scope on gB targets [34,52]. Previously, it has been shown by Brey et al. [34] that a very small amount of BiTE protein is needed for function; thus, we were confident that we would achieve killing abilities in the respective assays. Furthermore, we aimed to show different targets to potentially address different time points during infection and even allow combination therapy.

### 3.3. Design of a Nano-Luciferase-Release Assay

To prevent immune escape mechanisms of CMV from interfering with our assay readout, we built a system to prove BiTE efficacy. In addition to the glycoprotein-encoding vectors, every HEK293T cell line contained an nLuc-encoding vector under hygromycin selection. Since nLuc is produced intracellularly, these target cell lines can be applied for nLuc-release assays to evaluate the killing efficacy. Upon cell lysis, nLuc can be detected in the cell supernatant and serves to correlate the BiTE-mediated CB15 T cell-induced killing [50,57]. We opted for this readout to characterise BiTE efficacy and established the protocol within this study (Figure 2).

On day zero, 10,000 target cells are seeded in wells of a 96-well tissue culture plate in 100 µL of the medium without antibiotics (Figure 3). After 24 h, we added both 100 µL of the BiTE-containing supernatant and a 1:5 ratio of CB15 T cells matching the BiTE anti-CD3 domain. The co-culture was incubated under standard conditions for 24 h, the cell supernatant was harvested, and nLuc activity was measured using a luminometer. Experiments were performed as replicates of eight and repeated at least four times using the supernatant of independent BiTE products from independent transfections for each replicate. The results are visualised in Figure 7.

When analysing our data, it becomes apparent that all BiTEs show a significant increase in nLuc-release when co-incubated with T cells, indicating T-cell-mediated cell death (Figure 7). We normalised all measurements to our target and effector cell co-culture, to detect the effects of the BiTE addition. This allows us to assume efficacies for all of them. It is important to highlight that every replicate was produced using an individually prepared BiTE-containing supernatant. Since we chose not to isolate and standardise each BITE preparation, we want to underline that the assays showed significant results independent of a fixed concentration, with robust efficacy.

CB15 on target cells show an increased background signal compared to the mock groups, mainly due to their known increased level of cytokine secretion and autoactivation [53]. Nevertheless, using this standardised T-cell line allowed the circumvention of donor T cells’ variability, and this has previously been used in BiTE models [54]. By comparing them to our intervention group, we can highlight that CB15 autoactivation does not outcompete the actual abilities of the BiTE-mediated enhanced T-cell killing. In a majority of experiments, the CB15 activity was only slightly increased and comparable to a background cell lysis of BiTE only and mock groups.

## 4. Discussion

Considering advantages and disadvantages of current treatment options, this study analysed T-cell redirection for a major role in infection control. While we wanted to harness the efficacy of a personalised T-cell therapy, we also put an emphasis on off-the-shelf availability since patients at risk will benefit from immediate intervention and relatively manageable side effects. Potential high-risk patient groups are stem cell transplant recipients. During cytopenia, viral infection is not controlled and poses a lethal threat to already vulnerable individuals. We want to bridge the time until stem cell engraftment, approximately a month to a year. This is not only important with regard to infection control but also since CMV enhances transplant rejection and mimics/enhances graft-vs-host-disease (GVHD). GvHD is treated much differently from viral infection and would require the reduction in immunosuppression, completely contra-indicated in CMV disease. Since herpes viruses are notoriously difficult to eradicate, bridging therapy is the most reasonable option until physiological infection control by the adaptive immune system is (re)established.

Our main goal was to prove efficacy for bispecific T-cell engagers in the context of viral targets. During CMV infection, highly conserved glycoproteins are expressed on the surface of infected cells. Glycoprotein B and glycoprotein H are already a point of targeting in other treatments such as antibodies or even BiTE [52], while glycoprotein N is an emerging target and provides a different step into viral spread inhibition. The variety of targets additionally allows combination therapy to cover different replication cycle time points and avoid escape mutations.

We tested six engagers with six targets on three different glycoproteins, aiming at human T cells. While we want to move into animal models in the near future, this study focuses on the efficacy of the viral target single chain in vitro. The goal was to specifically evaluate new target binders for MCMV glycoproteins. Therefore, we used the human CD3ε-binding domain established in Blinatumomab as it is highly optimised and unlikely to interfere with our evaluation of the qualitative binding abilities of our novel viral scFv.

While both CMV and MCMV are susceptible to ganciclovir and mainly cause severe infection in immunocompromised hosts, they display strict species specificity. With very similar pathogenesis, we do find significant differences. Regarding the sequence and structure, the viral glycoproteins of CMV and MCMV are not entirely homologous. In total, previous research found up to 78 genes to be significantly homologous in both viruses, therefore making MCMV the established small-animal model for CMV infection [32,58,59]. We specifically focused on glycoproteins found in the murine CMV strain that are also similarly present in CMV to later allow an in vivo proof of concept of our novel BiTE.

Another approach would be to use CMV-specific glycoprotein targets and apply them to CMV-infected humanised mice. However, these models are difficult to establish and would be restricted to transferred human cell populations, and, therefore, they do not allow the studying of the viral spread or containment in vivo. Especially, in a model that is strongly dependent on the immune response, this important factor cannot be dismissed. Prospectively, for in vivo experiments, the use of half-life-extended BiTE (HLE) [60] would be beneficial in order to ease animal stress during experiments; conventional BiTEs with a short half-life would require frequent application or continuous infusion. However, despite several studies, no HLEs have been clinically approved, therefore making the canonical platform the better model.

Thus, the focus did not lie on comparing different constructs but on evaluating each one to increase our spectrum of options. However, for animal experiments, HLE BiTEs can easily be isolated using their silenced Fc domain and differentiated from serum protein due to their increase in size. We plan to investigate this in the future to establish murine CD3ε-directed HLE BiTE for in vitro and vivo analysis. The focus of this study, however, was to use anti-human CD3ε BiTE to establish our new binders and validate them.

We assume sufficient BiTE is present in the supernatant by validating the binding by flow cytometry before the nLuc release assays. This was previously shown by Dreier et. al. [54] where flow cytometry measured that the half-maximum mean in fluorescence was at least magnitudes above the half-maximum effective BiTE dose for cytotoxicity, leading to the assumption that very few molecules are necessary to trigger T-cell engagement mediated by BiTEs. In preparation of in vivo assays, it will be necessary to titrate BiTEs and investigate optimal dosage, which will be carried out with HLE BiTEs.

In this study, the nLuc release displayed some background signals in the control condition of the target cell co-incubation with CB15 T cells. We expected this from the usage of CB15 T cells that show some known activation [53,54]. However, the effect was not as potent as BiTE plus T-cell induced cell lysis and remained closer to spontaneous release for most experiments.

In the future, we will investigate the role of cytokine release after BiTE therapy in the murine infection model using MCMV. In other studies, it has been shown that Interferon-γ and tumour necrosis factor play a major role in infection reduction [61]. In CMV-infected cells in vitro, a major effect of BiTE efficacy relies on cytokine release instead of apoptosis induction [61]. For example, NK cells reduce viral spread measured by focus reduction, but their killing activity is antagonised efficiently by CMV [37]. This can be observed for both CMV and MCMV, as both rely heavily on Interferon γ release. While there is variety in the species-specific mechanism, cytokine release is integral across strains [38,39]. This contrasts with Blinatumomab, where the killing of tumour cells clearly results from T-cell activation [49]. In this mechanism also lies the risk of cytokine release syndrome that is observed in CD19 BiTE therapy with a large or central nervous system tumour burden [62,63,64]. However, we hypothesise that, with a comparatively lower number of infected cells, this will be easier to control in antiviral BiTE therapy. This promotes a big advantage of the BiTE, especially when compared to other immunotherapies such as CAR-T cells [65].

Zhou et al. [66] summarised the immunogenicity development in Blinatumomab and other bispecific reagents. In B-cell-directed BiTE, anti-drug antibodies (ADAs) were detected at very low titers, while, in clinical studies, novel BiTEs like Pasotuxizumab directed against prostate-specific membrane antigen (PSMA) showed high ADA development after s.c. administration but not prolonged i.v. infusion, with a contribution of T-cell epitopes in the drug [67]. Should this occur in our BiTE, we do not expect clinical relevance for this from our BiTE, as they are single-use bridging therapies and do not need multiple treatment cycles. Additionally, they would be used in immune-incompetent patients, likely with donor effector cells, leading to a low likability of ADA development.

In summary, this study successfully demonstrated the proof of concept for redirecting human T cells to viral glycoprotein targets. For this, we established a cellular model system encoding an nLuc reporter that can be modified for any surface protein to investigate T-cell-mediated target cell lysis. We found that BiTE proved a viable approach for the T-cell-mediated elimination of glycoprotein-expressing cells in vitro. We provide strong evidence that targeting cytomegaloviral gB, as well as the novel targets gH and gN, by using BiTE provides a feasible approach to successfully fight CMV in vulnerable patient cohorts.

## Figures and Tables

**Figure 1 viruses-16-00869-f001:**
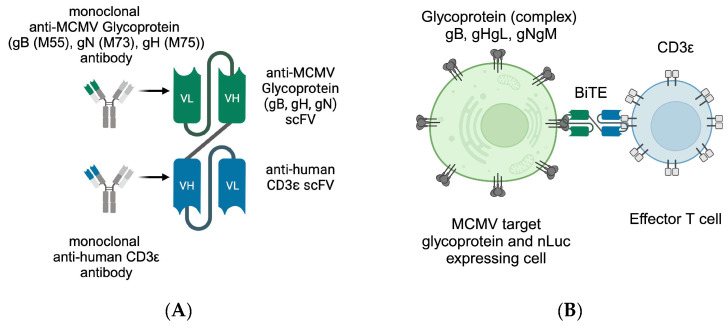
Schematic depiction of (**A**) the antibody-derived BiTE with MCMV glycoprotein recognising single-chain variable fragment (scFv) (green) and human CD3ε-binding scFv (blue), (**B**) the recognition principle of T cell (blue) and target cell (green, also expressing nLuc in our assays), with T-cell-receptor-independent CDε binding. This figure was created with BioRender.com (accessed on 3 May 2024).

**Figure 2 viruses-16-00869-f002:**
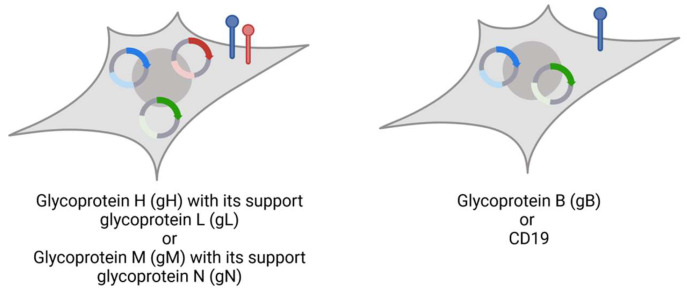
Stable cell lines expressing target (blue), supporter proteins (red), and nLuc (green). Glycoprotein H and N-expressing Human embryonic kidney (HEK) 293T cells with puromycin resistance are additionally transduced with gL or gM, respectively, linked with blasticidin resistance. Glycoprotein B and CD19 are expressed without co-glycoprotein. This figure was created with BioRender.com (accessed on 3 May 2024).

**Figure 3 viruses-16-00869-f003:**
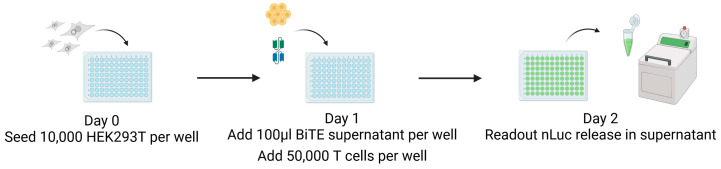
CB15 T-cell-induced killing: nLuc assay. On day zero, 10,000 target cells were seeded, and then, after 24 h, 100 µL of BiTE was added, plus T cells in a 1:5 ratio. nLuc activity as correlate for target cell killing was measured in cell supernatant after another 24 h using a luminometer. This figure was created with BioRender.com (accessed on 3 May 2024).

**Figure 4 viruses-16-00869-f004:**
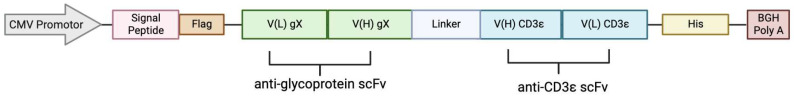
The design of the BiTE expression vector with CMV promotor, signal peptide, Flag-tag, protein cassette comprised of VL VH, a GSG linker, His-tag, and BGH poly A.

**Figure 5 viruses-16-00869-f005:**
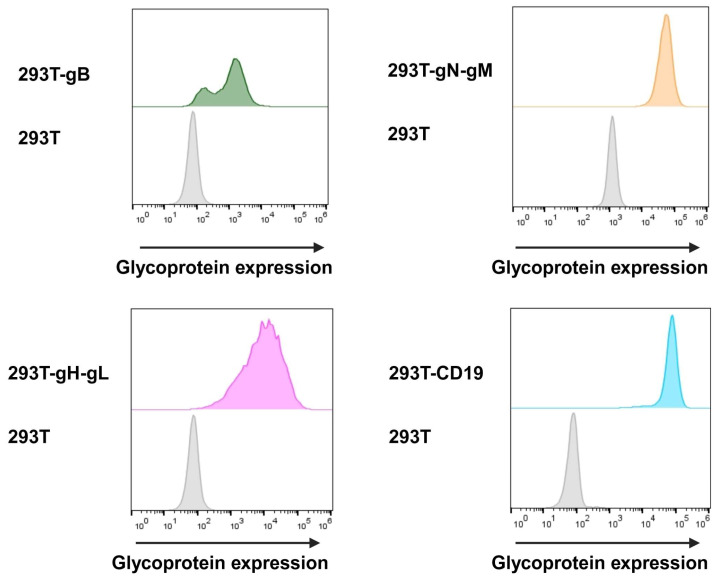
Target cells were surface-stained using hybridoma supernatant containing antibodies for gB and gH-gL, and anti-HA antibody for intracellular staining of gN-gM. These primary antibodies were then detected via secondary anti-mouse Alexa647-labelled antibody. Additionally, target cells expressing CD19 were stained using anti-CD19-Alexa647. As control, glycoprotein-negative HEK293T cells were stained identically.

**Figure 6 viruses-16-00869-f006:**
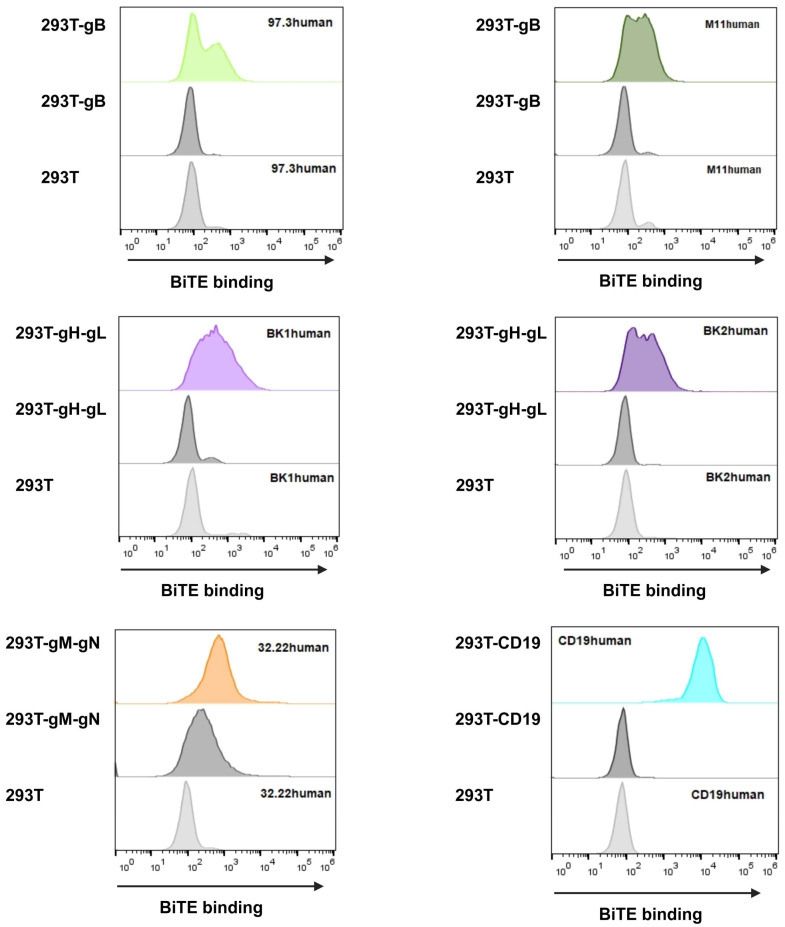
HEK293T MCMV-glycoprotein-expressing target cells were harvested and incubated in BiTE-containing supernatant for 1 h. BiTE protein bound to the respective viral glycoprotein were detected using anti-Flag antibody and anti-mouse Alexa647. Upper panel shows supernatant with BiTE on target-expressing cells; middle panel shows supernatant without BiTE on target-expressing cells; and lower panel shows supernatant with BiTE on HEK293T cells without target glycoprotein(s). The addition of BiTE is indicated by its respective name on the upper corner of the upper or lower panel.

**Figure 7 viruses-16-00869-f007:**
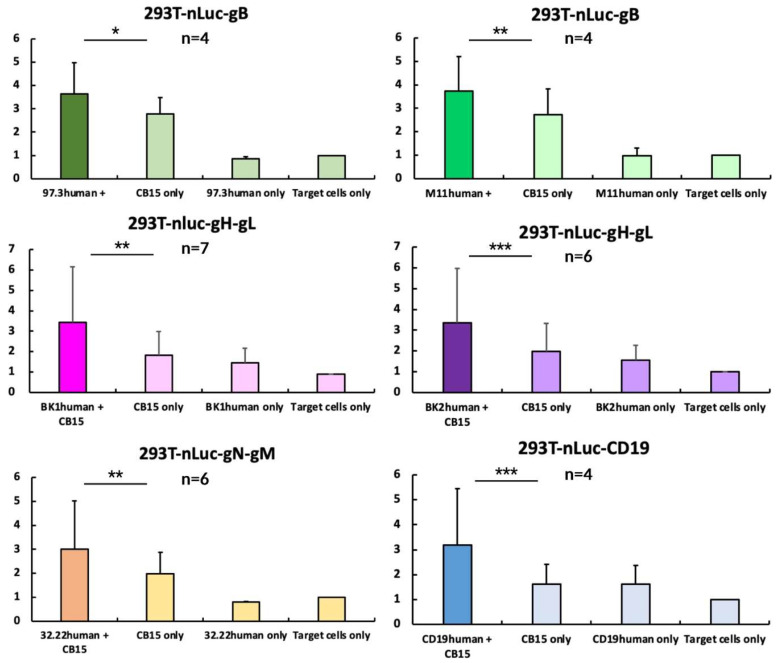
CB15 T-cell-induced killing assay. The indicated cell lines expressing viral glycoprotein complexes were incubated with CB15 T cells and BiTE. About 24 h later, supernatant was harvested and measured. For statistical analysis, measurements were normalised to nLuc-release upon addition of target cells alone and shown here via columns in mean (with 8 technical replicates per measurement). Error bars indicate standard error of mean. *p*-value was calculated using double-sided, unpaired Student’s t-test (* = *p* < 0.05; ** = *p* < 0.01; *** = *p* < 0.001). Exemplary individual measurements are depicted in Appendix A.

**Table 1 viruses-16-00869-t001:** BiTE plasmids.

Target	Hybridoma	Plasmid	Encoded BiTE
**gB**	97.3	pCDNA-97.3huCD3	97.3human
	M11	pCDNA-M11huCD3	M11human
**gN**	32.22	pCDNA-32.22huCD3	32.22human
**gH**	BK1	pCDNA-BK1huCD3	BK1human
	BK2	pCDNA-BK2huCD3	BK2human
**CD19**	HD37	pCDNA-CD19huCD3	CD19human

**Table 2 viruses-16-00869-t002:** Vector plasmids used for target cell lines with respective antibiotics for selection.

Plasmid	Resistance Marker
**pLV-gB**	Puromycin
**pLV-gN**	Puromycin
**pLV-gM**	Blasticidin
**pLV-gH**	Puromycin
**pLV-gL**	Blasticidin
**pLV-CD19**	Puromycin
**pLV-nLuc**	Hygromycin B

**Table 3 viruses-16-00869-t003:** Antibodies used in this study. Dilutions were used according to manufacturer’s protocol or 100 μL of supernatant per 96-well sample for hybridoma supernatant.

Antibody	Origin	Dilution	Assay
Anti-Flag^®^ BioM2	Mouse, #F9291, Merck	1:1000	BiTE detection gM detection
Anti-HA	Mouse, #901515, Biolegend (San Diego, CA, USA)	1:1000	gN detection
BK2 (Hybridoma)	Michael Mach, Virology Erlangen	100 μL hybridoma supernatant	gH detection
M11 (Hybridoma)	Michael Mach, Virology Erlangen	100 μL hybridoma supernatant	gB detection
anti-CD19-647	BioLegend, #363040	1:200	CD19 detection
anti-mouse IgG-647	Thermo Fisher Scientific, #A31571	1:1000	Secondary antibody

## Data Availability

The original contributions presented in the study are included in the article/Appendix A, further inquiries can be directed to the corresponding author/s.

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
