# Peer review of "Evaluation of Bispecific T-Cell Engagers Targeting Murine Cytomegalovirus"

_viruses, 2024, doi:10.3390/v16060869_

Round 1

Reviewer 1 Report

Comments and Suggestions for Authors

I have a tolerable knowledge of human and mouse CMV biology and cant understand this paper. The BiTE technology requires a diagram in the Introduction to explain the ligands. From the introduction, I cant see how antigen specificity is conferred.

There is also a logical flaw in assuming that the suppressive effects of CMV specifically target responses initiated by the MHC. Several mechanisms influence T-cells and IL-10 homologues affect most aspects of the immune system.

The evidence cited to dismiss NK cells (saying that they only act via IFN) is poor. Refs 34-36 are not the lest word on NK function in this context….and of course the NK responses to CMV are fundamentally different in humans and mice…and differ between mouse strains.

I believe gB, gN and gH are proteins of human CMV, with poor sequence similarity with their MCMV homologues. I also believe that HEK 293 T cells are human so it would be a strange combination to put them in a mouse... so  why is the strategy being tested with MCMV?

Perhaps C15 cells are murine but this isn’t stated….are they CD4 or CD8?....what is their specificity and origin?

The manuscript is full of abbreviations that I cant translate…only RPMI is defined.

The volumes and concentrations in Table 3 are not helpful…100ul in what?

I now come to the Results and some aspects become clearer, but...

The sentence “As control, we adapted the sequence of Blinatumomab targeting CD19 for human CD3e” is not clear….I thought this was a mouse system. Do human and mouse CD3e cross react?  If so this should have been explained in the strategy.

Lines 274-5 should also have been explained earlier (with Fig 1).

Figure 5 is incomprehensible…what is meant by “xxxhuman” on the 1st and 3rd plots?...the same codes appear in Fig 6.

The Discussion repeats the need to better CMV therapies for transplant recipients…this was provided earlier. It does not address why the induced T-cell responses would evade immunosuppression (induced via therapy or the virus itself). It has also confused me more re CB15 cells…human or mouse?

Line 415...273 cells grow rapidly in poor media. I doubt that CB15 competes for nutrients.

Comments on the Quality of English Language

The sentence structure can be improved but this isnt a big problem

Reviewer 2 Report

Comments and Suggestions for Authors

Menschikowski and colleagues provide a proof-of-concept study describing the use of bispecific TC engager constructs to target viral glycoproteins with the goal to kill MCMV infected cells. This technology is known and has been the subject of previous studies. While experiments on virus-infected cells are missing to fully convince me of the usefulness of this approach in the context of actual infection as here, most viral glycoproteins are present in very low numbers on the surface of an infected cell and immunoevasins are at play, the study does not overstate its results and clearly shows the usefulness of BiTE-mediated recruitment of TC to kill the selected target-expressing cells. However, I worry about the leakiness of the Luc-system and the power it provides regarding the readout. The effects might be understated by the chosen experimental system as they are generally rather weak. I also find some weak points in the experimental system regarding target expression and a suboptimal BiTE concentration which need to be clarified or addressed experimentally. Besides the seemingly weak effect on target cells, I see issues with the statistical analyses and suggest revision. Overall, the study is well-written, although wording seems a bit sloppy sometimes and the manuscript would benefit from a more concise style of writing.

Major comments:

1) Please elaborate on the immunogenicity of BiTE regarding potential in vivo approaches

2) Although it is clear that this is in preparation of MCMV-mouse in vivo experiments, a proof-of-concept study could also include paired experiments on HCMV glycoprotein expressing cells to validate the results in light of previous work. While it was mentioned that these studies have been conducted by others (ref 39), this can be discussed in more details to highlight the potential translational aspect more.

3) It would be interesting to co-express certain known immunoevasins although the used CB15 TC might be resistant.

4) CB15 cells seem to be highly reactive (autoreactive), also creating some background. Regarding point 3), is it possible to use primary mouse TC in this approach instead? Can the authors elaborate on why they did not use primary mouse TC on MCMV infected cells?

5) In Fig 6, I did not fully understand what 100% refers to here. Is it just cells+CB15? If so, the statistics do not work, as you only have one data point. Generally, it is counterintuitive to have percentages exceeding 100 as this makes no sense in the sense of the word. I would suggest decimals instead (fold change). As you generally reach inductions of only 0.5 to 2-fold, please elaborate on the effectiveness of the treatment. Does this kill all cells? You might want to set the killing of all cells to 1 (chemical killing) and no killing to 0 (no treatment) and then slot all the conditions including a +PMA control for reference. You term the untreated control “mock”, but this treatment is not a mock control of CB15 alone if I get it correctly. Did the “target cell alone” control receive BiTE? I would suggest to indicate the treatments differently so it becomes clear what went into the experiment.

6) Regarding statistics on Fig 6, you wouldn’t call it a “calculation of significance” as this implies you were calculating until the results reach significance. Also, as you are comparing groups, I would suggest ANOVA instead of t-test. I also suspect the error bars to be SEM instead of SD, please clarify.

7) The weak effects of TC-BiTE on target cells could be explained by low BiTE binding as seen in Fig 4/5. While you could argue that this recapitulates the low surface expression of CMV glycoproteins on infected cells, in a proof-of-concept study I would like to see more comparable and high target expression. You might consider sorting of the transduced cell lines to account for differences in BiTE binding. This way you could also make claims regarding the most potent BiTE+target pair.

8) Further, especially the 97.3 binding to gB-293T in Fig 5 seems to generate 2 populations which hints at an insufficient 97.3 concentration. This is also similar in all the gB and gH/gL targets. A titration experiment would clarify if the used concentration is too low or if you actually still have parental cells in your transductant culture, which seems to be the case at least in the 293T-gB culture (Fig 4). I would suggest sorting of target cells and a titration of BiTE on targets to find an optimal setup.

Comments on the Quality of English Language

needs some improvement regarding wording

Round 2

Reviewer 1 Report

Comments and Suggestions for Authors

The review has been answered well

Comments on the Quality of English Language

The language is OK

Author Response

We want to thank you again for your valuable input to enhance the quality of this manuscript.

Reviewer 2 Report

Comments and Suggestions for Authors

The authors have adressed all my concerns, revised parts of the manuscript, rearranged figures and clarified some points. I am generally satisfied, but one major point remains, which I think is very important (point 3).

1) The authors would like not to add controls or rework existing experiments. I understand that the scope of the study is explorative in a way and many controls I suggested would not change the impact of the study. Still, the overall quality of the experiments and therefore their persuasiveness is lacking imo. It is up to the editor.

2) The authors would like not to provide more experiments. I suspect that this study will be cited to go ahead with or publish the upcoming in vivo studies and should be serviceable to claim that this generally works, so I understand the motivation and see that the minimal requirement is met.

3) The authors stick with the statistics in Fig 7. However, as this is the key figure and should convince the reader of effectiveness, I would strongly suggest re-work or omit the statistics and change the presentation. I agree that ANOVA is not needed the way it is argued in the point-by-point response. However, I have major concerns regarding the statistical comparison to normalized data. If your main message is to compare CB15 only to CB15+BiTE and set all the CB15 only to 1, you are twisting the data for significance as there is no variance in the CB15 only data now. You should choose a different point of normalization (target only?!) or not normalize or not do statistics. Also, what you show are not biological replicates, but technical replicates and individual experiments. Biological replicates would require the use of primary cells or animals.

My suggestions would be:

- normalize to CB15 only but omit statistics

- normalize to target only and do statistics

Generally, I would say that in vitro assays do not need statistics at all but since the authors double down on this calculation and make this an argument for effectiveness, this has to be done correctly or omit statistics. Imo p-values have basically no meaning anyways. It doesn’t make a weak effect more believable or important in in vitro assays where you want to show reproducibility. Statistics are generally more suitable to in vivo analyses or if you would have used primary cells from different donors in the analysis. However, you use statistics to validate that across experiments you get the same result, which is generally not how you would want to use statistics. You could also say that the fact that you have to do that is an argument against the system. To me it only matters if the effect is visibly obvious and that there is not an unreasonable variance across experiments. To this point, SEM is also not suitable and I would show the range instead as you go for reproducibility and not “biological” variance. Again, showing individual measurements as dots is more what you would do with in vivo data. Here, bar graphs showing the inter-experimental means and range would be more suitable.

This whole point is not to annoy the authors, but vitally important as the entire paper hinges upon the differences in Fig 7 and it needs to be clear that the comparison between CB15 only and CB15+BiTE really is fair and pronounced enough to warrant further use of the system. You can use statistics to drive home the point that this works, but you would have to do it fairly, especially since the effect is weak overall and as some of the individual measurement actually show no effect at all (judged from the colored dots). I suspect that the CB15 only also shows a wide spread and you wouldn’t get low p-values because of it. Again, to me this is no problem as long as the effect is still visible. The p-values do not gloss over the fact that the effect is weak anyways.

Comments on the Quality of English Language

Generally well-written and easy to understand. Could be improved but does not prevent understandability.

Author Response

Response to Reviewer #2 (points raised in italics)

The authors have adressed all my concerns, revised parts of the manuscript, rearranged figures and clarified some points. I am generally satisfied, but one major point remains, which I think is very important (point 3).

We want to thank you again for your valuable input to enhance the quality of this manuscript.

  • The authors would like not to add controls or rework existing experiments. I understand that the scope of the study is explorative in a way and many controls I suggested would not change the impact of the study. Still, the overall quality of the experiments and therefore their persuasiveness is lacking imo. It is up to the editor.

Thank you for your understanding.

  • The authors would like not to provide more experiments. I suspect that this study will be cited to go ahead with or publish the upcoming in vivo studies and should be serviceable to claim that this generally works, so I understand the motivation and see that the minimal requirement is met.

Thank you for your understanding and input.

  • The authors stick with the statistics in Fig 7. However, as this is the key figure and should convince the reader of effectiveness, I would strongly suggest re-work or omit the statistics and change the presentation. I agree that ANOVA is not needed the way it is argued in the point-by-point response. However, I have major concerns regarding the statistical comparison to normalized data. If your main message is to compare CB15 only to CB15+BiTE and set all the CB15 only to 1, you are twisting the data for significance as there is no variance in the CB15 only data now. You should choose a different point of normalization (target only?!) or not normalize or not do statistics. Also, what you show are not biological replicates, but technical replicates and individual experiments. Biological replicates would require the use of primary cells or animals.

Thank you very much. We have revised toward a normalization to Target cells only and changed the nomenclature for the individual experiments.

My suggestions would be:

- normalize to CB15 only but omit statistics

- normalize to target only and do statistics

We have normalized to Target cells only.

Generally, I would say that in vitro assays do not need statistics at all but since the authors double down on this calculation and make this an argument for effectiveness, this has to be done correctly or omit statistics. Imo p-values have basically no meaning anyways. It doesn’t make a weak effect more believable or important in in vitro assays where you want to show reproducibility. Statistics are generally more suitable to in vivo analyses or if you would have used primary cells from different donors in the analysis. However, you use statistics to validate that across experiments you get the same result, which is generally not how you would want to use statistics. You could also say that the fact that you have to do that is an argument against the system. To me it only matters if the effect is visibly obvious and that there is not an unreasonable variance across experiments. To this point, SEM is also not suitable and I would show the range instead as you go for reproducibility and not “biological” variance. Again, showing individual measurements as dots is more what you would do with in vivo data. Here, bar graphs showing the inter-experimental means and range would be more suitable.

Thank you. We have edited Figure 7 to match your suggested profile. We decided to keep * to indicate significance between CB15 and CB15+BiTE for easier visualization; But we agree that diagrams can speak for themselves.

This whole point is not to annoy the authors, but vitally important as the entire paper hinges upon the differences in Fig 7 and it needs to be clear that the comparison between CB15 only and CB15+BiTE really is fair and pronounced enough to warrant further use of the system. You can use statistics to drive home the point that this works, but you would have to do it fairly, especially since the effect is weak overall and as some of the individual measurement actually show no effect at all (judged from the colored dots).

We decided that showing p-value for individual experiments would be confounding as they are technical replicates, in our opinion. The p-value in the individual experiments/across the (earlier versions) colour-matching dots was usually much lower (and significant in all measurements), however putting all diagrams as shown in Figure S6 would have taken up too much space.

I suspect that the CB15 only also shows a wide spread and you wouldn’t get low p-values because of it.

CB15 only did not show high intra-experimental varieties per individual measurement. If we could put error bars on the dots, this group would actually have the smallest variance.

Again, to me this is no problem as long as the effect is still visible. The p-values do not gloss over the fact that the effect is weak anyways.

Thank you again for the comments and suggestion.